# The Intricate Epigenetic and Transcriptional Alterations in Pediatric High-Grade Gliomas: Targeting the Crosstalk as the Oncogenic Achilles’ Heel

**DOI:** 10.3390/biomedicines10061311

**Published:** 2022-06-03

**Authors:** Paul Huchedé, Pierre Leblond, Marie Castets

**Affiliations:** 1Childhood Cancers & Cell Death (C3), Université Claude Bernard Lyon 1, INSERM 1052, CNRS 5286, Centre Léon Bérard, Labex DevWeCan, Centre de Recherche en Cancérologie de Lyon (CRCL), 69008 Lyon, France; paul.huchede@lyon.unicancer.fr (P.H.); pierre.leblond@ihope.fr (P.L.); 2Department of Pediatric Oncology, Institut d’Hematologie et d’Oncologie Pédiatrique, Centre Léon Bérard, 69008 Lyon, France; 3Translational Research Pole in Pediatric Oncology, Centre Léon Bérard, 69008 Lyon, France

**Keywords:** pediatric glioma, HGG, epigenetics, H3K27M, H3G34R, transcriptional networks, developmental programs, cell of origin, targeted therapy, clinical management

## Abstract

Pediatric high-grade gliomas (pHGGs) are a deadly and heterogenous subgroup of gliomas for which the development of innovative treatments is urgent. Advances in high-throughput molecular techniques have shed light on key epigenetic components of these diseases, such as K27M and G34R/V mutations on histone 3. However, modification of DNA compaction is not sufficient by itself to drive those tumors. Here, we review molecular specificities of pHGGs subcategories in the context of epigenomic rewiring caused by H3 mutations and the subsequent oncogenic interplay with transcriptional signaling pathways co-opted from developmental programs that ultimately leads to gliomagenesis. Understanding how transcriptional and epigenetic alterations synergize in each cellular context in these tumors could allow the identification of new Achilles’ heels, thereby highlighting new levers to improve their therapeutic management.

## 1. Introduction

Tumors of the central nervous system (CNS) are the most common cause of cancer-related deaths in children aged 0 to 14 years [1]. Among these, pediatric high-grade gliomas (pHGGs) account for 15–20% of all brain tumors in children and are characterized by a poor outcome [1,2]. No effective therapy has so far been identified, and the 2-year survival rates thus range from 10% to 30% for supratentorial pHGGs and are less than 1% for diffuse intrinsic pontine glioma (DIPG), a particular subgroup of diffuse midline gliomas (DMGs) arising in the pons [2,3,4].

Specific studies have been conducted on these pediatric pathologies and have led to the conclusion that gliomas in children are different from those arising in adults [5]. Indeed, like most other childhood tumors, the mutational burden of pHGGs (comprising DMGs) is lower than that in adult cancers [6,7]. They often display key oncogenic mutations, such as lysine(27)-to-methionine and glycine(34)-to-arginine/valine substitutions in histone 3 (H3K27M and H3G34R/H3G34V, respectively) [8,9,10,11] or fusions in receptor tyrosine kinases (RTKs) such as NTRK, ALK, ROS1, or MET [12,13,14]. A lot of key genetic drivers of pHGGs induce major chromatin reorganization: pHGGs are then more and more appreciated as epigenetic malignancies, with subsequent revision of their classification according to this criterion [15].

Current treatments depend on tumor localization and on the age of the child. For patients with pHGGs localized outside the brainstem, a complete resection of the tumor is performed if feasible, followed by focal radiation therapy combined most often with temozolomide treatment according to the scheme published by Stupp et al. in adults with high-grade gliomas [16]. pHGGs of the midline are most often not accessible to surgery. The standard treatment is therefore based on exclusive radiotherapy, or with concomitant temozolomide or other targeted therapies depending on clinical studies. This was, for example, the strategy of the BIOMEDE study, which proposed a combination of radiotherapy with erlotinib, dasatinib or everolimus, depending on the tumor profile concerning EGFR, PDGFR and PTEN. However, the results of this study remain disappointing, since the median overall survival (OS) is barely 1 year for patients included in the best treatment arm [17]. For the youngest patients, and patients who are unable to receive radiation therapy, alternative chemotherapy regimens can be proposed [18,19].

In this review, we present an inventory of key molecular events associated with pHGGs occurrence and their importance in the stratification of the disease, focusing particularly on epigenetic components. We describe how the disruption of the epigenetic landscape in a specific permissive transcriptional context ultimately leads to pediatric gliomagenesis. In addition, we propose that transcriptional network alterations reflect the co-option of developmental processes and act in an intricate interplay with epigenetic rewiring to promote oncogenic properties. Finally, we illustrate the clinical relevance of this epigenetic/transcriptional crosstalk (Figure 1).

## 2. Epigenetic Remodeling at the Root of pHGGs Etiology

pHGGs are a heterogeneous group of malignancies with different molecular etiologies. Stratification of pHGGs according to transcriptomic, genomic and epigenomic similarities is essential for the establishment of a robust molecular classification and the design of effective and relevant stratified treatments.

### 2.1. Chromatin Reorganization Due to Genetic Events as Oncogenic Drivers in pHGGs

Unlike the multiple genomic events that accumulate to induce tumorigenesis in many adult cancers, a special feature of childhood cancers is epigenetic disruption, causing a massive dysregulation of gene expression [20]. Two main mutational events responsible for this broad epigenetic reorganization in pHGGs are the p.Lys27Met and p.Gly34Arg/Val substitutions in histone 3 (H3K27M and H3G34R/V, respectively) [8,9,10,11]. These two crucial residues, located within the highly conserved N-terminal tail of the protein, influence the dynamic regulation of chromatin structure and accessibility to transcriptional activator or repressor complexes [21]. Around 80% of DIPGs harbor the H3K27M mutation, and 20% of pHGGs located on the cerebral hemispheres harbor H3G34R/V mutations [22], making them major oncogenic events in these diseases.

On one hand, the H3K27M mutation is more frequently found in the *H3-3A* (*H3F3A*) gene, encoding histone variant H3.3, or in the related *H3C2* (*HIST1H3B*)—and to a lesser extent *H3C3* (*HIST1H3C*)—genes, encoding histone variant H3.1 [8,9,10,11,22]. Surprisingly, these mutations are only present on a minority of the total tumor histones pool, with only 5 to 10% of histones affected. This proportion is nonetheless sufficient to cause a global depletion of the repressive H3 lysine 27 trimethylation (H3K27me3) mark [23,24,25]. This has been attributed to H3K27M, which acts as a dominant negative mutation, due to its strong inhibitory affinity for the methyltransferase enzymatic subunit of the polycomb repressive complex 2 (PRC2) EZH2, consequently abrogating the ability of PRC2 to establish H3K27me3 repressive chromatin domains [23,24,25]. This H3K27me3 loss is concomitant to the increase in H3 lysine 27 acetylation (H3K27Ac) [23,26], a marker of active chromatin and transcription throughout the genome [23,24,26,27]. However, this global effect hides a more complex mechanism by which EZH2 remains active at very specific loci and causes a punctual increase of H3K27me3 [24,25,26,27], sparing the strongest PRC2 targets [28,29]. Last, inhibition of H3K27 trimethylation occurs only when H3K27M mutated histones are deposited in chromatin, suggesting that EZH2 is inhibited only when chromatin patterns are being duplicated in proliferating cells [30]. These studies highlight a new avenue of research into H3K27M biology, considering the importance of quiescent cells in promoting oncogenic properties and treatment resistance in gliomas [31,32,33].

On the other hand, H3G34R and H3G34V mutations, exclusively found in the *H3-3A* (*H3F3A*) gene, were reported to alter the active H3 lysine 36 trimethylation (H3K36me3) mark [23,34]. Unlike H3K27M, only H3G34R/V mutated histones show an alteration of H3K36me3. It was first proposed that this key mark decreased due to H3G34R/V methyltransferase SETD2 inhibition [23]. However, in specific genomic regions, it was shown that expression of H3.3G34R at endogenous levels in mouse embryonic stem cells results in augmentation of H3K36me3 and H3K9me3 through the inhibition of KDM4A/B/C, resulting from its higher affinity for these three dual H3K9/H3K36 demethylases [34]. Moreover, G34 mutations promote PRC2 activity, thereby increasing H3K27me2/3, by blocking SETD2-mediated H3K36 methylation at active enhancers [35]. Interestingly, the correction of the mutated H3G34R allele in patient-derived tumor cells indicates that the mutation incorporates at already highly expressed genes. This suggests a fine regulation mechanism rather than genome-wide chromatin alterations seen in its K27M counterpart [36,37]. Given the opposing effects of H3K9me3, H3K27me3 and H3K36me3 in transcription regulation [38,39,40,41,42,43,44], and since H3K36me3 alteration is a major feature of certain cancers [45,46,47], further studies are required to fully elucidate the mechanism by which H3G34R/V mutations rewire some key sites of the epigenome.

### 2.2. Establishment of a New pHGGs Classification: Epigenetic as a New Guide

In the fifth edition of the WHO classification of tumors of the CNS, published in 2021, the value of molecular diagnostics in pHGGs tumor classification has been put forward [15]. This new classification distinguishes “Pediatric-type diffuse high-grade gliomas”, which is subdivided into 4 subgroups with distinct molecular and biological features, and additional distinctions in anatomical location, age at diagnosis and overall survival, namely: diffuse midline glioma, H3K27-altered; diffuse hemispheric glioma, H3G34-mutant; diffuse pediatric-type high-grade glioma, H3-wildtype and IDH-wildtype; and infant-type hemispheric glioma. Of note, the H3K27-altered subgroup also comprises the newly defined EZHIP-overexpressing DMGs subgroup, mimicking H3K27me3 loss observed in H3K27M tumors [48].

H3K27-altered tumors are specific to midline structures, divided between the pons and non-brainstem midline regions. They display a significantly shorter time to death from disease, with a median OS of 11 months. Conversely, H3G34R/V tumors are almost entirely restricted to the cerebral hemispheres and have a longer median OS of 18 months. Of note, the age of patients is significantly different depending on the underlying mutation, varying from a median of 5 years for H3.1K27M tumors to 7 years for H3.3K27M and 15 years for H3.3G34R/V tumors [22]. This sequential occurrence of H3-mutated tumors likely pinpoints the importance of the ontogenic context in the expression of the oncogenic potential of each mutation, a point that will be detailed later in this review.

Regarding infant-type hemispheric gliomas, they have paradoxical clinical behavior whether found in children or adults. Indeed, low-grade tumors have a higher mortality rate, while high-grade tumors have a better outcome (5-year OS of 54.5%), complicating the association of histology and outcome in infants [12,13]. Their overall survival is relatively good compared with that of older children with pHGGs. This clinical observation may be linked to a more differentiated state of high-grade tumors after chemotherapy [12] and highlights the fact that the strong plastic potential of pediatric tumors can be exploited by treatments. Moreover, major genetic fusion events present in this subgroup, involving receptor tyrosine kinases *ALK*, *ROS1*, *NTRK1/2/3* and *MET* genes [12,13,14], are found almost exclusively in the high-grade group of infant-type glioma. The mechanism at play that leads to excessive proliferation arises from a classical constitutive activation of the Ras/Raf/MEK/ERK pathway, notably by aberrant ERK1/2 phosphorylation [12,13,14]. Of note, ALK fusion is sufficient to drive the infant form of HGGs with 100% penetrance when electroporated in utero at E14.5 in a mouse model, while tumor formation is rare when the fusion gene is expressed postnatally [13]. This advocates for a prenatal origin of this subtype and highlights the importance of the epigenetic context in tumorigenesis. The definition of this full-fledged entity constitutes a breakthrough for the clinical management of this molecularly unique subgroup.

Mutations in *IDH1*, which are frequent in adult gliomas, are only found in a very small proportion of pHGGs [22]. The BRAF V600E mutation is observed in around 6% of both midline and hemispheric pHGGs and is associated with a better prognosis [22,49,50]. These two events are not currently associated with a particular pHGGs subgroup.

Finally, the histone H3-wildtype and IDH-wildtype entity has heterogeneous characteristics. Improvement of its clustering was achieved via inclusion of DNA methylation patterns, which allowed the identification of 3 subgroups, with the worst prognosis attributed to the MYCN subgroup (median OS 14 months), followed by the RTK1 subgroup (median OS 21 months), and a better survival for the RTK2 subgroup (median OS 44 months) [51]. Aside from presenting different methylation profiles, each subgroup displays enrichment of different gene amplifications, namely *PDGFRA* amplification in the RTK1 subgroup and *EGFR* amplification in the RTK2 subgroup, to which homozygous deletions of *CDKN2A/B* and losses involving Chr10q can be added [51]. Interestingly, it was reported that H3.3G34R/V also upregulates *MYCN* through H3K36me3 binding, illustrating potent non-exclusive oncogenic mechanisms between pHGGs subgroups [37]. The molecular and clinical characteristics of each subgroup are detailed in Table 1.

The development of ultra-high throughput and scalable tools to analyze chromatin, as well as investigations into DNA methylation and modifications in histone profiles, will undoubtedly constitute routine decision-making tools to guide optimal management of patients with pHGGs.

## 3. Synergy between Transcriptional and Epigenetic Rewiring in pHGGs: A Matter of Oncogenic Window

### 3.1. Oncogenic Contribution of Non-Genetic Ontogenic Factors in Gliomagenesis

pHGGs exhibit a clear spatio-temporal and -molecular pattern of incidence. As mentioned above, H3K27M and H3G34R mutations are mutually exclusive and have a distinct anatomical distribution within the CNS. Tumors of the thalamus, brainstem and spinal cord frequently exhibit the K27M mutation and occur in younger children, whereas the G34R mutation is found exclusively in tumors of the cerebral lobes and occurs mainly in adolescents and young adults [11,22].

Pediatric tumors develop in the context of actively growing tissues. Thus, this peculiar ontogenic environment may be subverted to promote malignancy, resulting in a unique spectrum of tumors that differ greatly from those of adults [52]. However, in the case of pHGGs, there seems to be a delay between the postnatal developmental window during which brain structures reach their maturity and the peak of pHGGs occurrence. For example, pons development is achieved within the first five years of life and hardly evolves afterwards, and Ki67-positive (a marker of proliferation) cells become rare and barely change from 1.5 years of age in this structure [53]. The pons-proliferative phase thus differs from the peak of incidence of H3K27-altered tumors, which occurs around 6–7 years [22]. Similarly, the brain hemispheres have almost reached their adult size at the age of 10 years [54], while the incidence of H3G34R/V tumors peaks at 15 years of age [22].

However, the precise pattern of gliomagenesis may match developmental waves of myelination in the human CNS [20,55,56]. In particular, glioma cells likely take advantage of the secretion of trophic factors, such as brain-derived neurotrophic factor (BDNF) and neuroligin-3 (NLGN3), which are regulated by neuronal activity. Hence, ontogenic events necessary to structure the neural network could also support pHGGs occurrence [57,58,59]. Along the same lines, it is tempting to speculate that spatio-temporal patterns of occurrence of pHGGs may result from hijacking of developmental pathways such as FGF, WNT, Notch and BMP signaling [60,61], the activation of which oscillate during pre- and postnatal development in a tightly regulated spatio-temporal equilibrium.

Thus, taking the ontogenic context in which pHGGs appear into consideration is undoubtedly a key factor in understanding the contribution of non-genetic mechanisms to the striking pattern of occurrence of these tumors.

### 3.2. Importance of the Cell-of-Origin in the Activation of Oncogenic Transcriptional Networks

In addition to the impact of the environment, the state of the cell in which oncogenic mutations occur plays a key role in tumorigenesis. Beyond improving our knowledge of pHGGs biology, defining the identity of the lineage or cell(s) at the origin of the different subgroups is a key clinical issue. Indeed, if the epigenetic context clearly participates in defining the transformation capacity of a cell, it also constitutes an important determinant of resistance and treatment response [62,63,64].

Through scRNA-seq technology, developmental cell states and their cooperation have been decrypted in both H3K27M and H3G34R/V tumors. Fresh tissues from diagnostic biopsies of H3K27M tumors allowed the identification of four different trajectories, one related to cell cycle, and the other three to differentiation states, namely astrocyte-like, oligodendrocyte-like, and oligodendrocyte precursor cell-like (OPC-like). This highlights a putative developmental hierarchy in which proliferated OPC-like cells are the main population, which both self-renew and give rise to the AC-like and OC-like cells (Figure 2) [65]. These data suggest, in line with anterior studies reporting the identification of this unique population of immunophenotypic neural precursor cells of the brainstem [66,67], that H3K27M-tumors may originate from a precursor of the OPC lineage.

This view has been challenged using several models based on the induction of H3K27M expression in different neural progenitors. Expression of H3.3K27M together with TP53 inactivation and PDGFRA amplification results in glioma-like tumors when targeted to mouse postnatal neural progenitor cells (NPCs) [68,69] or xenografted human embryonic stem cell (hESC)-derived NPCs [70]. Similarly, conditional expression of H3.3K27M in nestin-positive cells, in a TP53 KO and PDGFRA-constitutively active context leads to brainstem gliomas from neural stem and progenitor cells [71]. Whereas expression of H3.3K27M postnatally fails to trigger tumor development, its electroporation in NPCs at embryonic day E12.5–E13.5 induces diffuse tumors when H3.3K27M, combined with Trp53 loss, is expressed permanently [23,72]. Recently, Haag and colleagues developed a human-induced pluripotent stem cell (iPSC) model carrying an inducible H3.3K27M allele in the endogenous locus to study the impact of the mutation on several precursors of the oligodendrocytic lineage, from iPSCs to NSCs and OPCs [73]. Interestingly, only NSCs gave rise to tumors upon induction of H3.3K27M and TP53 inactivation in an orthotopic xenograft model, arguing in favor of the crucial role of cell state in the expression of the oncogenic potential of pHGGs’ mutations. Moreover, H3.3K27M induction in NSCs was shown to lead to sustained expression of stemness and proliferative genes and a premature activation of OPC programs that may synergistically cause tumor initiation [73], reconciling this finding with the OPC-like signature found in patient samples [65]. Lastly, these NSCs resemble embryonic neuroepithelial-like NSCs [74] rather than later NPC developed in previous models [68,69,70], which is consistent with previous results suggesting a prenatal appearance of the mutation (Figure 2).

Patients’ tumors bearing H3G34R/V harbor a forebrain cortical interneuron lineage transcriptomic signature, including radial glia, neuronal progenitor, and prenatal interneuron gene programs [75,76]. Interestingly, introduction of the H3.3G34R mutation in neural progenitor cells of the developing ventral forebrain derived from hESCs is sufficient to form tumors that recapitulate key features of H3G34R/V patient tumors when combined to the double loss of TP53 and ATRX [22,75]. Moreover, scRNA-seq analysis revealed a dual neuronal and astroglial identity, strikingly devoid of oligodendroglial programs [76], highlighting a major difference between H3K27M and H3G34R/V tumors. Even if the proportion of each cellular subtype varies greatly from one patient to another, these results show another unique developmental hierarchy, involving interneuron progenitors that differentiate into both astrocytes and neurons within H3G34R/V tumors. As cortical interneurons are generated during embryonic development in transient progenitor domains of the ventral telencephalon [77,78], this strongly suggests a prenatal origin of H3G34-mutated pHGGs (Figure 2). Despite these advances, no models of tumor initiation are currently available. Their development would therefore be of major interest to decipher biological mechanisms at the roots of H3G34-mutant pHGGs.

The importance of the cellular state in which the mutation occurs may go beyond cell type. Indeed, recent work indicates that refinement can go as far as to induce a difference in behavior depending on the spatial identity of a given progenitor. Bressan and colleagues showed that the phenotypic impacts of H3K27M and H3G34R are different in engineered human fetal NSC cultures arising from distinct brain regions [36]. On the one hand, H3K27M only exerts oncogenic activity in hindbrain NSCs by increasing both proliferation and clonogenicity. On the other hand, H3G34R has no oncogenic activity on either of the two spatially distinguished fore- and hindbrain NSCs. However, it triggers a strong cytostatic effect on the latter, suggesting a tolerability and ability to further evolve towards a potent oncogenic transformation in forebrain NSCs; accordingly, increased proliferation and clonogenicity was observed in these cells when H3G34R was combined with PDGFRA overexpression and TP53 knock-out [36].

These recent insights suggest that expression of H3K27M and H3G34R/V mutations requires a precise permissive transcriptional context enabled by both a specific cell differentiation state and regional identity. Whereas G34-mutated gliomas may be neuronal malignancies of forebrain interneuron progenitors stalled in differentiation, K27-altered gliomas may arise from hindbrain NSCs, where stemness programs are maintained concomitantly with a premature activation of OPC programs. In both cases, and despite the obvious differences in age of incidence between these two subgroups, growing evidence suggests a prenatal origin of epigenetically disrupted pHGGs, where histone 3 mutations remain indolent until further oncogenic signals intervene [36].

### 3.3. Hijacking of Transcriptional Developmental Pathways and Maintenance in an Immature Epigenetic State as the Core of pHGGs

As mentioned above, H3G34-mutated tumors are more likely to reinforce or stabilize a specific forebrain regulatory circuit already present in the cell of origin. Using a cell line model corrected for the H3G34R mutation, significant downregulation of many genes associated with forebrain development and neuroprogenitor proliferation were observed [36], including CDK6, SOX1/2, POU3F2/3, ARX, and DMRTA2 [79,80,81,82,83,84]. Moreover, engineered fetal forebrain NSCs leading to glioma-like cells (gathering H3.3G34R, TP53 KO, and PDGFRA overexpression) were shown to upregulate key transcription factors involved in neuroprogenitor self-renewal and proliferation, such as OLIG2 and SOX3, and in forebrain-specific markers, including DMRTA2, EMX2, NR2F1 and HIVEP2 [82,85,86,87,88]. An essential transcription factor for neuronal differentiation, ASCL1 [89], is downregulated by the combination of H3G34R, TP53 and ATRX mutations in a forebrain-specified ESC model [75]. Meanwhile, genes associated with stem cell maintenance are also upregulated, including NOTCH1, NOTCH2 and NOTCH2NL ligands, as well as key target genes of the same pathway, HES1 and HEYL [75,90,91,92,93,94].

Interestingly, Notch pathway activation has also been observed in H3K27-altered pHGGs. Indeed, induction of both H3G34R and H3K27M mutations in healthy astrocytes and H3 wild-type pediatric glioma cells activates Notch signaling, in particular through an increase in the expression of NOTCH1, HES5 and ASCL1 genes [95]. Of note, in these models, ASCL1 expression is paradoxically reported as being up or downregulated by H3 mutations, suggesting that this is rather its expression alteration that destabilizes the differentiation process [75,95]. Increased transcription of these genes is in part due to the recruitment of both H3K36me3 and H3K27ac at the corresponding promoters. Moreover, restoration of the H3 wild-type form in H3.3K27M DIPG cell lines by gene editing results in downregulation of these genes through an increase in the H3K27me3 mark of their corresponding promoters [95]. Co-option of the Notch developmental pathway then seems to be a key oncogenic mechanism in pHGGs through the maintenance of early neural precursor stem and proliferative properties. This aberrant activation of such developmental pathways strongly relies on the peculiar epigenetic context triggered by H3K27M or H3G34R.

Indeed, it has been demonstrated that H3.3K27M preserves H3.3 genome-wide distribution when introduced in fetal hindbrain NSCs, leading to a similar phenotype as H3G34R tumors, namely H3.3K27M drives tumorigenesis by locking tumor-initiating cells in their pre-existing epigenomic state [96]. The majority of H3.3K27M localizes at active enhancer and promoter regions. Surprisingly, even if the mutation causes a global demethylation and increased acetylation, as previously described, H3.3K27M leads to focal H3K27ac loss, decreased chromatin accessibility and reduced transcriptional expression at active enhancers of genes involved in neural differentiation, such as SOX9 [82], and genes involved in neurodevelopmental diseases, such as CHN1, CTNND2 and NGFR [97,98,99]. However, to a lesser extent H3.3K27M also binds to PRC2-bound regions and notably decreases key neural markers, including DLX1, DLX2, DLX3 and NEUROG2/NGN2 [100,101,102].

Differences in activation of transcriptional programs have been described between the two main H3K27M isoforms, H3.1K27M and H3.3K27M, which could be related to epigenetic modifications driven by each histone. Indeed, gene enrichment profiling shows a strong enrichment for the oligodendrocytic or proneural-glioblastoma multiforme (GBM) signatures in H3.3K27M tumors, whereas H3.1K27M tumors are enriched in astroglial or mesenchymal GBM signatures [103]. However, scRNA-seq shows no differences in astrocytic-specific genes between H3.1 and H3.3K27M cells, which suggests that this signature may be linked to microenvironmental cells [104]. Indeed, H3.1K27M is distributed across the genome, whereas H3.3 is enriched at active regulatory elements, leading to different chromatin accessibility and transcription factor (TF) binding profiles between H3.3K27M and H3.1K27M tumors [104]. H3.3K27M tumors are enriched in TFs involved in early neural development, included several genes of the RFX family [105], or POU5F1, encoding the well-known OCT4 TF involved in cell pluripotency [106]. Moreover, a subgroup of H3.3K27M tumors preferentially activates enhancers associated with noncanonical WNT signaling as well as increased expression of WNT planar cell polarity (WNT/PCP) pathway members [104]. When exposed to WNT5A, H3.3K27M, but not H3.1K27M cells, were shown to undergo a rapid extension of cytoskeleton-containing neurite-like processes, promoting oncogenic properties through increased cell viability and the formation of gap junction-coupled tumor microtubes [104], likely reinforcing their resistance to therapies [107,108]. Accordingly, mutations of WNT pathway members have been observed in rare cases of pHGGs—such as AMER1 and APC mutations—, and this particular group seems to be associated with a poorer survival [22]. Hijacking of WNT signaling, a key actor of early development [109,110,111,112], could thus specifically support the acquisition of oncogenic properties in H3.3K27M tumors.

Differentially accessible enhancer elements in H3.1K27M tumors are enriched in NF-𝜅B, EGF-activated receptor and MAPK pathways, without strong impact of H3.1 mutation on activation of specific developmental programs [104]. However, in a very interesting way, 85% of H3.1K27M tumors are also mutated on the ACVR1 gene, encoding the BMP receptor ALK2 [22,113,114,115,116]. These mutations, also present in a rare genetic disorder characterized by progressive heterotopic ossification—fibrodysplasia ossificans progressiva (FOP) [117]—trigger both enhanced responsiveness to BMP signals and autophosphorylation of the receptor independently of its ligand binding. This results in higher SMAD1/5/8 phosphorylation [61,118,119,120,121], the main effectors of the canonical BMP pathway, and increased expression of key target genes of the ID family [61,120,121]. Dissection of the synergy between H3.1 and ACVR1 mutations was investigated in different models. Mice expressing the recombinant ACVR1 G328V allele in Olig2-positive cells developed neurological anomalies, due to the arrest of oligodendroglial lineage cell differentiation, as evidenced by the decreased expression of oligodendrocyte maturation markers [121]. When combined with H3.1K27M and PIK3CA H1074R, an additional lesion found in ACVR1-mutated tumors [122,123], high-grade diffuse gliomas occurred. Moreover, when ACVR1 R206H was induced in nestin-positive-TP53 KO pontic neurospheres, it cooperated with H3.1K27M in promoting DIPG pathogenesis via activation of Stat3 signaling and upregulation of mesenchymal markers such as CD44 or TNC [120], thus reinforcing results obtained by Castel and colleagues [103]. However, it was only in the presence of the PDGFA ligand that tumor incidence increased and median survival decreased in a orthotopic xenograft mouse model [120]. This is all the more surprising, since PDGFRA amplification is a common feature of H3.3K27M but not H3.1K27M tumors [8,11,22,124]. However, in the previously described ACVR1-^floxG328V/+^; Olig2^Cre/+^ model, PDGFRA was significantly upregulated following ACVR1-mutant expression, suggesting an alternative mechanism to the one found in H3.3K27M tumors [121]. Aberrant PDGF activation also holds true for H3G34R/V tumors, since PDGFRA mutations are found in 50% of tumors, where its aberrant expression is further amplified by the recruitment of H3K27Ac and GSX2-associated cis-regulatory elements on its promoter [76]. Concomitantly and similarly to ACVR1-mutated cells, increased ID1 levels have also been observed in forebrain-specified H3G34R ESCs [75]. Considering the role of ID1 and PDGFRA in NSC maintenance and OPC identity [125,126,127,128,129,130], similar core transcriptional network alterations can be proposed, as they constitute a potential common oncogenic mechanism in pHGGs. Moreover, the presence of other alterations of the BMP pathway—such as BMP2K, BMP3 mutations or ID2, ID3 amplifications—[22], in addition to the above-described ACVR1 mutation, highlight the importance of the hijacking of the BMP signaling pathway in these tumors.

Altogether, the concept of an oncogenic cooperation is undoubtable in the context of pHGGs. There is a clear combination between epigenomic disruption and alterations of developmental pathways, mirroring specific developmental windows [131]. This is illustrated by several combinations of epigenetic, transcriptomic and genetic events, which involve different mutations on histone 3, the maintenance of key developmental networks hindering NSC differentiation and activation of key signaling pathways—namely Notch, WNT, BMP and PDGFR—whose co-option, in the context of pHGGs, promote oncogenic properties. This oncogenic state could be maintained by a crosstalk between the permissiveness of the epigenetic state, allowing the expression of transcriptional signaling pathways, and the concomitant regulation of this epigenetic state by some of these developmental pathways [132,133,134,135,136]. These biological insights must be considered before designing new targeted and efficient treatments (Figure 1).

## 4. Targeting the Synergistic Epigenetic/Transcriptional Oncogenic Node: A Path towards New Therapies

The therapeutic management of pHGGs faces two challenges: the intrinsic complexity of these tumors, also resulting from their location, and the impossibility of transposing the results obtained in adults, given their specificities in children. As an example, the alkylating agent temozolomide has demonstrated a significant anti-tumor effect in adult patients with HGGs and is now used as a standard-of-care with radiation therapy [16]. Conversely, it exhibits no clinical benefit for pediatric DMGs [137,138,139]. As previously suggested by Filbin and Monje [20], the unique features of pHGGs cells could be particularly permissive to certain treatments, which are under exploration (for detailed review on current clinical trials involving the use of targeted therapies for pHGGs, see Findlay et al. [140]). A first promising way towards innovative therapies is the use of histone deacetylase (HDAC) inhibitors such as panobinostat, showing preclinical activity against DMG H3K27-altered cells in vitro and in vivo on patient-derived xenograft models [141,142,143]. As the HDACi-induced partial rescue of cell phenotype could only be transitory [144], a combinatorial and synergistic approach of therapeutics should be anticipated, notably to target the oncogenic cooperation between epigenetic alterations and hijacked transcriptional developmental pathways.

In this respect, the impact of oncohistones on transcriptional networks reveals sensitivities to compounds that target oncogenic mechanisms rather than the mutation itself. The use of isogenic models is crucial to ensure treatment specificity to oncogenic targets activated by the mutation. In such models, expression of H3.3K27M was shown to modify sensitivity to therapeutic compounds, depending on the cell context in which it is induced. For example, while the H3.3K27M-induced Res259 pediatric low-grade glioma cell line is more sensitive than the parental isogenic cell line to eleven drugs—including the multi-tyrosine kinase inhibitors dasatinib and midostaurin, the MEK inhibitor trametinib or the bromodomain inhibitor OTX015—the sensitivity of two other pHGGs cell lines to these drugs remains unchanged upon H3K27M expression [145]. These results seem to corroborate previous studies in the field [146,147,148], and such isogenic models will thus be of interest to evaluate the pan-tumors/more restricted potential of new drug combinations. Along these lines, a cross-comparison of isogenic pediatric glioma cell lines, in which H3K27M or H3G34R was corrected/introduced, revealed the potential of targeting the Notch pathway in H3-altered pHGGs. Indeed, when using the γ-secretase inhibitor DAPT, which blocks NOTCH1 cleavage, all DIPG cell lines corrected for H3.3 exhibited increased IC50 compared to their mutated counterparts, alone or in addition with irradiation treatment [95]. H3.3K27M induction shows a drastic decrease in the IC50 of DAPT only in astrocytes, whereas a similar effect was observed after H3.3G34R induction in a pediatric glioma cell line [95]. Of interest, the combination of the MRK003 Notch inhibitor with the BET bromodomain inhibitor JQ1 shows a synergistic efficacy and increased apoptosis compared to monotherapies in DIPG cells [149]. This illustrates the interest of targeting an oncogenic cooperation between Notch signaling and epigenetic mechanisms.

Similar approaches were conducted in a non-tumoral background that more accurately mimicked the milieu in which mutations occurred. Primary human NSCs in which the K27M mutated or the wild-type forms of histone 3 were induced showed no difference when treated with molecules targeting transcriptional activators such as bromodomain and extraterminal domain (BET) family members, cyclin-dependent kinase 7 (CDK7) and CBP/p300, which could have been promising strategies considering transcriptional dependencies in the disease [141].

Consistently, and with the aim of targeting an oncogenic cooperation, the new generation of HDACi with dual properties could address the transient effect and lack of efficacy of HDACi monotherapy and has generated interest in recent years in cancer treatments [150]. A CRISPR screen revealed that knockout of *KDM1A*, encoding lysine-specific demethylase 1 (LSD1), sensitizes DIPG cells to HDACi [151]. Treatment of these cells using the bifunctional molecule Corin, a dual HDAC and LSD1 inhibitor, alters the global patterns of histone modifications of DIPG cells, resulting in cell cycle arrest, increased toxicity in vitro, and decreased tumor size in an orthotopic mouse xenograft model [151]. Moreover, Corin treatment induces differentiation through upregulation of genes involved in neurogenesis and by decreasing progenitor markers in vitro and in vivo. The panel of dual HDACi thus offers new opportunities in H3-altered pHGGs therapy, potentially combined to cyclin-dependent kinases (CDKs) [152], PI3K [153] or receptor tyrosine kinases (RTK) inhibitors [154,155,156], which are signaling pathways of interest in these tumors (reviewed by Duchatel et al. [157]).

Further strategies are currently being tested and are particularly promising, including selective dopamine receptor D2 (DRD2) antagonists ONC201 and ONC206 [158], which are also potent agonists of the mitochondrial caseinolytic protease P (ClpP) [159,160]. They show a strong anti-tumoral activity in vitro and in vivo through both TRAIL induction and AKT/ERK inhibition [161,162]. Moreover, following activation by ONC201, ClpP drives degradation of mitochondrial respiratory chain enzymes, engaging a p53-independent apoptosis [163]. Of interest, and considering glioma cells’ plasticity, this treatment also triggers a lineage shift from a proliferative, oligodendrocyte precursor-like state to a mature, astrocyte-like state [164]. ONC201 treatment is currently in clinical trials and seems to be an effective therapeutic approach [165,166,167,168], and its combination with other targeted therapies may be of particular interest.

## 5. Conclusions and Future Perspectives

Although pHGGs undoubtedly remain a major therapeutic challenge in pediatric oncology, major advances have been made in recent years in understanding their molecular etiology. The importance of both the epigenetic component and the crosstalk between chromatin rewiring and the hijacking of developmental transcriptional pathways have been highlighted by several innovative works. These discoveries offer three major perspectives for improving the care of children and young people harboring these diseases: in the short term, inclusion of the methylome analysis to the wide spectrum of omics analyses performed at diagnosis has demonstrated its robust potential to define homogeneous molecular groups of pHGGs. In the medium term, characterization of oncogenic mechanisms, particularly epigenetic ones, has revealed new therapeutic possibilities, which are currently in pre-clinical or clinical phases of development, as part of precision medicine approaches. In the longer term, a key challenge will likely be to imagine new therapeutic levers precisely targeting the synergistic signaling nodes between epigenetic and transcriptional alterations, in particular by transposing their oncogenic potential from the processes at play during embryonic development.

To be considered by the field, and as immunotherapies might be a new treatment option for several pediatric malignancies [169,170,171,172,173] and adult gliomas [174,175,176], it has recently been shown that H3K27M-cells show a high expression of the disialoganglioside GD2, whereas H3-wildtype pHGGs cells expression of GD2 was far lower. The development of anti-GD2 CAR T cells demonstrated a robust anti-tumoral effect of H3K27M pHGGs cells in vitro and in vivo on patient-derived orthotopic xenograft models [177]. The response of the first four patients appears very promising [178]. One of the challenges in the coming years will be to explore the impact of the singular epigenetic context of H3-altered pHGGs, and its dialogue with transcriptional signals, on the response to immunotherapies.

## Figures and Tables

**Figure 1 biomedicines-10-01311-f001:**
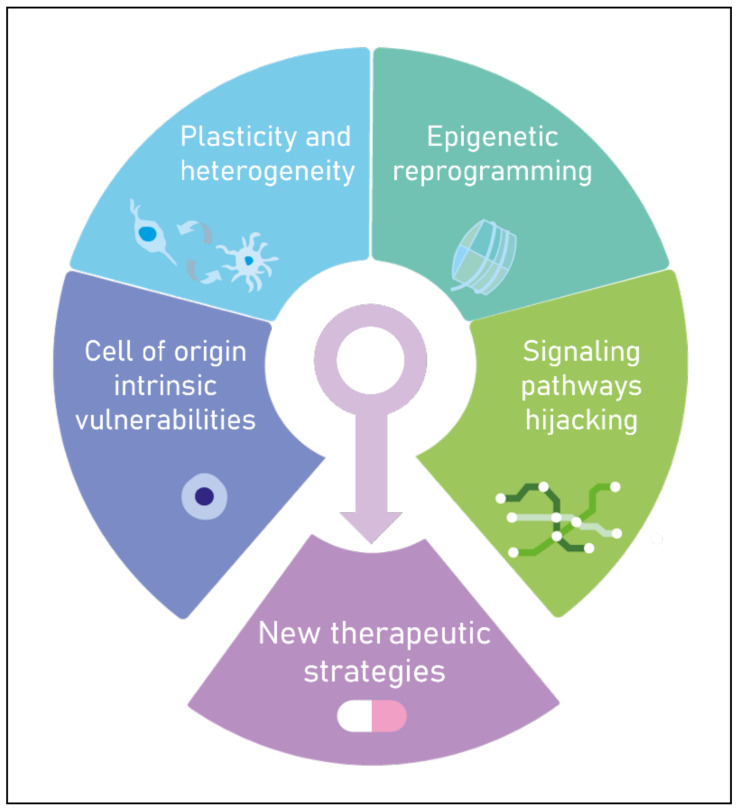
Elaboration of new efficient therapeutic strategies for pediatric high-grade gliomas will require the integration of the cell of origin intrinsic vulnerabilities, plasticity and heterogeneity, epigenetic reprogramming and signaling pathways hijacking.

**Figure 2 biomedicines-10-01311-f002:**
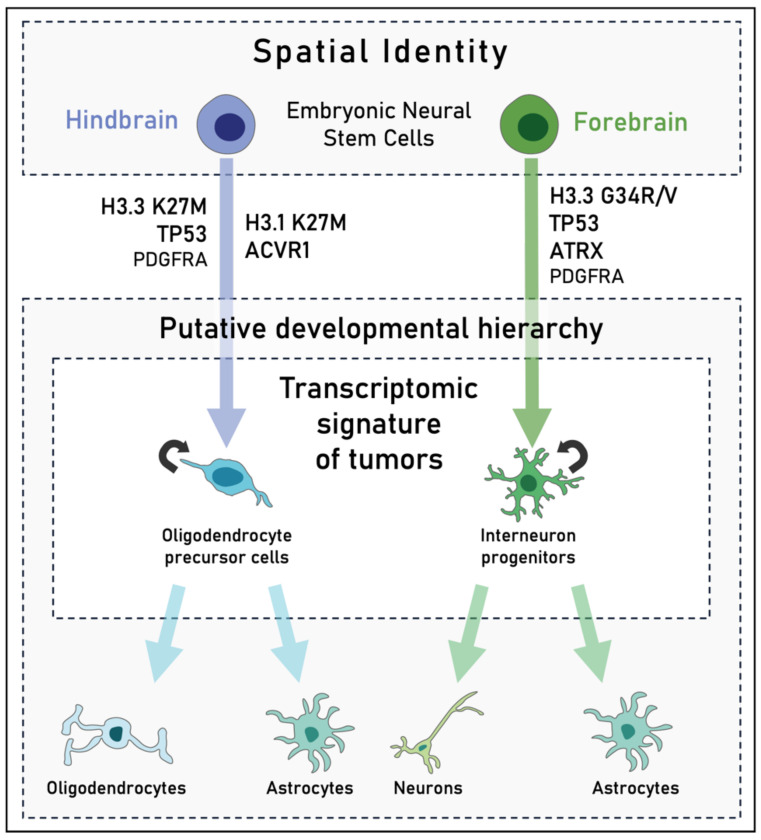
Schematic representation of H3-altered pediatric high-grade gliomas genesis and tumor hierarchy. H3 K27M tumors may arise from embryonic neural stem cells (NSCs) of the hindbrain. They have a cycling oligodendrocyte precursor cell (OPC)-like transcriptomic signature, with smaller subpopulations resembling differentiated oligodendrocytes and astrocytes. H3 G34R/V tumors may also arise from embryonic NSCs but of the forebrain, with a proliferative population of interneuron progenitors that would form both neurons and astrocytes. These two tumor subtypes recreate a putative developmental hierarchy that mirrors both glial (K27M) and neuronal (G34R/V) lineages.

**Table 1 biomedicines-10-01311-t001:** Molecular and clinical characteristics of pediatric type diffuse high-grade gliomas.

Name of the Tumor Entity	Major Event	Associated Events	Age at Diagnosis [Years]	Median OS [Months]
Diffuse midline glioma, H3K27-altered	H3.3K27M	TP53, PDGFRA	7	11
	H3.1K27M	ACVR1, PIK3CA	5	15
	EZHIP overexpression	ACVR1, PIK3CA	10	16
Diffuse hemispheric glioma, H3G34- mutant	H3.3G34R	TP53, ATRX, PDGFRA	15	18
	H3.3G34V			
Diffuse pediatric-type high-grade glioma, H3-wildtype and IDH-wildtype		MYCN		14
	ø	PDGFRA	10	21
		EGFR, CDKN2A/B		44
Infant-type hemispheric glioma	ALK, ROS1, NTRK1/2/3	ø	0.23	23
	or MET fusions			

## Data Availability

Data sharing not applicable.

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
