# Peer review of "The Intricate Epigenetic and Transcriptional Alterations in Pediatric High-Grade Gliomas: Targeting the Crosstalk as the Oncogenic Achilles’ Heel"

_biomedicines, 2022, doi:10.3390/biomedicines10061311_

Round 1

Reviewer 1 Report

The review is interesting and well written.

Author Response

We thank the reviewer for his/her supportive comments.

Reviewer 2 Report

Huchedé and collaborators provide an original reivew  describing the molecular specificities of pediatric high-grade gliomas (pHGGs) subcategories in the context of epigenomic rewiring caused by Histone 3 mutations and their implications in the orign of these tumours and how this knowledge can help to provide new therapies. The manuscript is concise and well written. The authors may consider some aspects that could improve the quality and impact of the manuscript:

1.- In the introduction, the authors may add a couple of senetnces about the current treatments for pHGGs and whether there are significant differences in treatment response depending on the histone mutational profile. This information could be added just before citing Table 1 (line 159).

2.- Line 398, (REF) should be substituted by the correct reference.

3.- Figure 2. The authors may consider the symbol that illustrates "Signaling pathways hijacking" This arrow does not fit very well with the rest of the iconography.

4. In conclusions and Future Perspectives section, the authors may want to add some info related to the implication of these epigenetic rewiring and the possibility that these molecular changes could contribute to immune evasion and/or the sensitivity/resistante to the immunotherapies that are unde development.

Author Response

We thank the reviewer for his/her supportive comments.

We now have improved the quality of the ms following his/her recommendations:

  1. Few sentences related to current treatments have now been added in introduction (L46-58) ;
  2. Reference has been added L410.
  3. Symbol has been changed on Figure 2.
  4. Few sentences related to immunotherapies' potential have now been added at the en of the conclusion.

Reviewer 3 Report

The ms by Huchede et al. is an elegantly written narrative review highlighting an important, but still omitted, relations between epigenetic background of the disease (HGG in that specific case), and transcriptional/translational aspects. It gives a nice summary of the current knowledge in that scoped area.

I would have few suggestions to consider:

1. Table 1 - slight reformatting: "Name of the tumor entity" should probably be enough since the paper is submitted in 2022, therefore WHO 2021 is obvious, also [yrs] and [mths] in the title line - then no repetitions in every line; also names of entities - the cover 2 lines anyway - maybe it could be distributed into the lines not to divide the names?

2. Figure 2, in my opinion, is a perfect graphical abstract, has not a lot to do with the ms text itself in the place where it is situated.

Author Response

We thank the reviewer for her/his supportive comments.

We have modified the ms according to her/his advices :

  1. Table 1 has been modified.
  2. We modified the position of Figure 2.